# The Effect of Long COVID-19 Infection and Vaccination on Male Fertility; A Narrative Review

**DOI:** 10.3390/vaccines10121982

**Published:** 2022-11-22

**Authors:** Soheila Pourmasumi, Alireza Nazari, Zahra Ahmadi, Sophia N. Kouni, Cesare de Gregorio, Ioanna Koniari, Periklis Dousdampanis, Virginia Mplani, Panagiotis Plotas, Stelios Assimakopoulos, Christos Gogos, Georgios Aidonisdis, Pavlos Roditis, Nikos Matsas, Dimitrios Velissaris, Gianfranco Calogiuri, Ming-Yow Hung, Servet Altay, Nicholas G. Kounis

**Affiliations:** 1Social Determinants of Health Research Center, Rafsanjan University of Medical Sciences, Rafsanjan 7717933777, Iran; 2Clinical Research Development Unit, Ali-Ibn Abi-Talib Hospital, Rafsanjan University of Medical Sciences, Rafsanjan 7717933777, Iran; 3Department of Surgery, School of Medicine, Rafsanjan University of Medical Sciences, Rafsanjan 7717933777, Iran; 4Pistachio Safety Research Center, Rafsanjan University of Medical Sciences, Rafsanjan 7717933777, Iran; 5Speech Therapy Practice, 26221 Patras, Greece; 6Department of Clinical and Experimental Medicine, University of Messina Medical School, 98122 Messina, Italy; 7Department of Internal Medicine, Division of Cardiology, University Hospital of South Manchester NHS Foundation Trust, Manchester M23 9LT, UK; 8Department of Nephrology, Saint Andrews State General Hospital, 26221 Patras, Greece; 9Intensive Care Unit, Patras University Hospital, 26500 Patras, Greece; 10Department of Speech Therapy, University of Patras, 26500 Patras, Greece; 11Department of Internal Medicine, Division of Infectious Diseases, University of Patras Medical School, 26500 Patras, Greece; 12COVID-19 Unit, Papageorgiou General Hospital, 56403 Thessaloniki, Greece; 13Interbalkan Medical Center, 57001 Thessaloniki, Greece; 14Department of Cardiology, Mamatsio Kozanis General Hospital, 50100 Kozani, Greece; 15Cardiology Private Practice, 30131 Agrinion, Greece; 16Department of Internal Medicine, University of Patras, 26500 Patras, Greece; 17Pneumonology Department, Civil Hospital “Ninetto Melli”, Pietro Vernoti, 72027 Brindisi, Italy; 18Department of Internal Medicine, Immunology and Infectious Diseases, Section of Allergology and Clinical Immunology, University of Bari Medical School, 70121 Bari, Italy; 19Division of Cardiology, Department of Internal Medicine, Shuang Ho Hospital, Taipei Medical University, New Taipei City 23561, Taiwan; 20Division of Cardiology, Department of Internal Medicine, School of Medicine, College of Medicine, Taipei Medical University, Taipei 110, Taiwan; 21Taipei Heart Institute, Taipei Medical University, Taipei 110, Taiwan; 22Department of Cardiology, Faculty of Medicine Trakya University, Edirne 22030, Turkey; 23Department of Internal Medicine, Division of Cardiology, University of Patras Medical School, 26500 Patras, Greece

**Keywords:** infertility, COVID-19, sperm, SARS, vaccine

## Abstract

Earlier research has suggested that the male reproductive system could be particularly vulnerable to SARS-CoV-2 (COVID-19) infection, and infections involving this novel disease not only pose serious health threats but could also cause male infertility. Data from multi-organ research during the recent outbreak indicate that male infertility might not be diagnosed as a possible consequence of COVID-19 infection. Several review papers have summarized the etiology factors on male fertility, but to date no review paper has been published defining the effect of COVID-19 infection on male fertility. Therefore, the aim of this study is to review the published scientific evidence regarding male fertility potential, the risk of infertility during the COVID-19 pandemic, and the impact of COVID-19 vaccination on the male reproductive system. The effects of COVID-19 infection and the subsequent vaccination on seminal fluid, sperm count, sperm motility, sperm morphology, sperm viability, testes and sex hormones are particularly reviewed.

## 1. Introduction

Infertility constitutes an important problem for families around the world [1,2]. Infertility can create social, economic and psychological distress to infertile patients, also affecting the economic burden ofhealth and clinical systems [3,4]. Based on the World Health Organization (WHO) definition, when couples have unprotected sexual intercourse for at least 1 year and pregnancy does not occur, they are categorized in the infertility group [5]. It is estimated that about 8–12% of couples suffer from infertility problems [6]. Infertility is due to male and female disorders and both genders are involved in infertility. It is estimated that in all infertility cases worldwide, about 50% of infertility is related to male factors and 50% to female factors [7]. A Global Burden of Disease (GBD) analysis reported that over the last three decades, infertility prevalence has increased annually in both men and women (∼0.3% vs. 0.4% respectively) [8].

Male infertility can result from three main parts: (A) testicular part, (B) ejaculated part and (C) hormonal part. Approximately, 50% of all male infertility is caused by abnormal sperm production (defects in spermatogenesis process) and defective sperm parameters (count, motility and morphology) [9,10]. These parts can be affected by several factors, including systematic disease, infections, medications, occupational parameters, psychological factors, lifestyle, etc. Consequently, these factors can have negative effects on male reproductive potential [11].

Since 2020, COVID-19 has rapidly spread across the world and infected hundreds of millions of people globally. This was an unknown virus and the experience of the scientific community concerning both short-term and long-term complications of COVID-19 infection continues to improve [12]. Laboratory assessments confirmed that COVID-19 has 46 different varieties and is detectable in human and animal body tissue and secretions [13].

COVID-19 infection can induce a variety of manifestations in the human body via the attack on respiratory, cardiac, immunology, digestive, urinary and neural systems. The effects of COVID-19 on fertility potential at the start of virus spread were unexpected and mysterious. In the second peak, several studies were designed to determine the effects of COVID-19 on fertility, confirming that this virus can also affect male fertility [14].

When the human body is infected by a virus, infection might further induce immune system response against the virus. An important signof infection is spiked temperature (>38 °C), which can have further harmful effects on the male reproductive system andsperm quality. It has been established that 4 to 11 weeks post fever, the quality of sperm decreases and the number of impaired sperm in seminal fluid increases. Based on scientific reports, three mechanisms may be responsible for negative COVID-19 effects on the male reproductive system: First, high testis temperature post fever is a defective factor for germ cell lines (spermatogonia cell line); Second, fever can affect other cells in the testis and seminiferous tubules (Sertoli and Leydig cells) and consequently create defects in exocrine and endocrine functions; Third, fever can decrease accessory gland secretion and this might be harmful for sperm viability and quality [15,16].

Both human genders are susceptible to COVID-19 infection, but it has been shown that men are at higher risk of infection, and the rate of infected men with COVID-19 is higher than in women [17].

Hence, infertility and its treatment have been affected during the COVID-19 pandemic, as in the first peak there was a lack of knowledge about the COVID-19 transmission process and its appearance in human reproductive secretions; and also, in different countries, infertility clinical centers were completely closed.

A study reported that COVID-19 infection was harmful for reproductive potential in male patients, affecting sex hormone level, testicular integrity and seminal quality [18].

After the COVID-19 pandemic and the death of millions of people around the world, serious efforts were made globally to produce a vaccine against COVID-19 [19]. For the first time, a Pfizer vaccine received emergency use approval by the Food and Drug Administration (FDA) and was rapidly administered for the reduction of infection severity [20]. The safety of the COVID-19 vaccine was identified by several laboratory tests, and the clinical and self-reports of cases who received both the first and second dose of the vaccine were evaluated [21,22].

In spite of no confirmed reports on the results of the vaccine on human physiology systems, there was public fear about the potential adverse effects of the vaccine on both the male and female reproductive systems [23]. Based on several fertility and reproductive reports, COVID-19 vaccines demonstrate no harmful effects on fertility, but to date many reproductive-age people are reluctant to receive the vaccine [24,25].

Therefore, in this study, we aimed to review scientific evidence about male fertility potential, the risk of infertility during the COVID-19 pandemic, and the effects of COVID-19 vaccination on the male reproductive system. Given the global burden of the COVID-19 pandemic, there is tremendous interest in understanding the potential health disparities in the offspring of men with a history of COVID-19. The effects of COVID-19 infection on male fertility are shown in Figure 1.

## 2. Search Methods

We searched related keywords in Medline (PubMed) based on the study purpose. The keywords were SARS-CoV-2 OR COVID-19, AND male infertility, OR sperm, OR sex hormone, AND testis OR orchitis, AND seminal fluid, OR fertility potential, AND COVID-19 vaccine OR SARS-CoV-2 vaccination. Studies reporting the effect of either SARS-CoV2 infection or COVID-19 vaccination on qualitative factors such as demographic parameters and quantitative markers, such as seminal fluid quality, sperm count, sperm motility, sperm morphology elements, sperm viability, testicular and sex hormone function, were analyzed.

## 3. SARS-CoV2

SARS-CoV-2 caused the COVID-19 disease pandemic at the end of 2019 [26]. There are several symptoms associated with COVID-19 infection, including a cough, gut dysfunction, shortness of breath, and fever. Furthermore, 80% of patients after infection with SARS-CoV-2 reported approximately mild symptoms, 5% of them severe, and death can occur in 3% of cases [27]. The COVID-19 epidemic has been associated with infertility [28]. Recent reports have confirmed that the COVID-19 virus is harmful for sperm parameters, sex hormone levels and fertility potential. There is considerable evidence reporting over 25 viruses, such as Herpes Simplex Virus (HSV) and Human Immunodeficiency Virus (HIV), that can pass the genital tract and negatively impact spermatogenesis [29,30,31].

### Possible Mechanisms of SARS-CoV-2 Affecting Male Reproduction

How can a virus cause damage to the reproductive system? This is an important scientific question and several hypotheses have been raised to answer this question. The main proposed mechanisms by which a virus impairs fertility include: (I) The virus can attack the spermatogonia germ cell line directly and impair sperm production; (II) The virus can spread in reproductive fluids and be transmitted by sexual intercourse; (III) The virus can induce inflammation in the reproductive tract. This inflammation can cause tissue damage, cell damage and sex hormone secretion damage in testis; (IV) The virus can cause male accessory gland infection, where secretion is an important substance to maintain male fertility potential; (V) The virus is a cause of infection-induced inflammation response and indirectly exerts negative effects on male fertility via the elevation of oxidative stress in the human body and the increase of reactive oxygen species; (VI) Finally, virus infection can trigger fever, and high body temperature could prove harmful for the reproductive system [32]. In conclusion, the inflammation chain can create damage to the body, and perhaps some of these mechanisms overlap.

## 4. COVID-19 Vaccination and Male Reproductive Potential

At the present time, limited studies have assessed the effects of the COVID-19 vaccine on male fertility factors including sperm parameters, hormonal level and testis impairment. In the scientific reports, researchers have confirmed that vaccination against COVID-19 did not exert harmful effects on male fertility. In a recent study, Adamyan et al. assessed two prior vaccines [33] (Pfizer-BioNTech and Moderna) on sperm parameters before and after the first and second vaccination dose in reproductive-age individuals. They reported no significant reduction in sperm analysis parameters. Olana et al. evaluated the effects of COVID-19 vaccination on male fertility parameters. In this study, no significant differences in spermatozoa parameters before and after vaccine inoculations were found. Furthermore, oxidative stress analysis, the activity of the cell membrane, and IL-6, as a marker of inflammation, were not increased in human seminal fluid by the mRNA COVID-19 vaccine [34]. Zhu et al. assessed COVID-19 vaccination impact on sperm count and motility. The study showed that there was no decrease in sperm count and volume post COVID-19 vaccination in comparison with the respective values before vaccination [35]. Barda et al. studied the relation between first and second dose of the vaccination and sperm concentration. Sperm concentration count was significantly increased after the second dose of the vaccine [36]. Chatzimeletiou et al. investigated the effects of the COVID-19 vaccine on sperm parameters in ankylosing spondylitis patients., revealing that the sperm count was significantly increased from the first to the third vaccine dose [37]. Olana et al. evaluated the effects of COVID-19 vaccination on male fertility parameters, reporting that there was no significant difference in oxidative stress level and sperm motility before and after vaccination [34]. In another study, Zhu reported that COVID-19 vaccination has no negative effects on sperm motility [35]. In a study with patients suffering from ankylosing spondylitis, COVID-19 vaccination could improve sperm motility after the second dose [37]. Kumar and his colleagues, in their study, demonstrated that there was no scientific evidence to confirm the impairment effects of COVID-19 vaccination on sperm parameters [38]. Sperm morphology was assessed in ankylosing spondylitis patients by Chatzimeletiou et al. as well. They reported that COVID-19 vaccination had no negative effects on sperm morphology; although the shape of sperm in ankylosing spondylitis individuals showed more defects, reporting that vaccination can exert positive effects on sperm morphology [37]. The effects of COVID-19 vaccination on testis tissue were assessed by Carto et al. [39]. They investigated orchitis and epididymitis in vaccinated and non-vaccinated cases. Their results showed that the rates of orchitis and epididymitis were significantly lower in vaccinated cases compared to non-vaccinated ones, suggesting that a single dose of COVID-19 vaccine can significantly decrease the risk of inflammation in testis.

According to a recent report, COVID-19 vaccinations are safe for the male reproductive system despite the fact that post the second dose about 15% of vaccinated individuals showed mild fever as a clinical symptom [40]. Consequently, vaccination was not able to promote a severe inflammatory response in human reproductive system. To date, no scientific data have been presented to support probability of testis tissue damage and sperm impairment after vaccination. Moreover, in a data analysis from an internet-based pregnancy study that follows U.S. and Canadian couples trying to conceive over time [41], the researchers followed more than 2000 women, 21–45 years old, from December 2020 to November 2021. All were trying to conceive without fertility treatments. Participants answered questions on income, education, lifestyle, and reproductive and medical histories. The latter included questions about COVID-19 vaccine status and positive SARS-CoV-2 tests. The women also provided information about their partners’ vaccine status and positive tests. Male partners aged more than 21 years answered similar questions. Female partners completed follow-up questionnaires every 8 weeks for up to 12 months or until they became pregnant. It was revealed that the chances of conception did not change with either partner’s vaccination status. The researchers adjusted for several factors that could influence the results. These included vaccine type, nationality, occupation, and history of infertility. The final conclusion was that the COVID-19 vaccination in either partner is unrelated to fertility among couples.

## 5. COVID-19 in Seminal Fluid

Major concerns about the impact of SARS-CoV-2 infection on male fertility have increased, and one of the important questions is the present probability of COVID-19 in the seminal fluid of infected individuals. For the detection of virus co-expression, both angiotensin I-converting enzyme 2 (ACE2) and transmembrane serine protease TMPRSS2 were required in the testis of COVID-19 individuals, raising concerns regarding the virus entrance from the blood–testis barrier and the presence of COVID-19 in testis tissue and seminal fluid. To date, different reports regarding the controversy of the presence of COVID-19 in seminal fluid have been published.

Some studies have reported the significant impact of COVID-19 infection on seminal fluid quality [42,43].

A report by Li et al. demonstrated that semen can shed the virus during the recovery phase, even though COVID-19 does not affect the semen of all men who have the virus. The seminal fluid of 15 cases with severe infection was tested and COVID-19 was detected in about 27% of patients [44].

Gacci et al. [43] showed that after recovery from COVID-19, 25 out of 43 male patients were diagnosed as oligocrypto-azoospermic, while 8 and 3 patients revealed azoospermia and oligospermia, respectively. Furthermore, IL-8 was found in the semen fluid of 76.7% of patients [43].

In a study published in 2020, researchers suggested that COVID-19 may be detectable in acute infected patients after the treatment and recovery period. They also discussed that COVID-19 is detectable in patients who have no infection symptoms [45]. COVID-19-infected patients demonstrated common infection symptoms including fever and inflammation. Fever is an important parameter that has negative effects on seminal fluid and quality of sperm. Low seminal and sperm quality post severe fever seem to continue for at least 90 days [46]. Furthermore, inflammation can activate the immune system and consequently the seminiferous tubules in testis affected by COVID-19, representing a main factor of seminal and sperm quality impairment [47].

In contrast to the above reports, several studies confirmed that there was no COVID-19 virus detection in the seminal fluid of infected patients. For example, Song et al. reported that the presence of COVID-19 in the seminal fluid of male patients post recovery could not be identified [48]. Furthermore, in another study, the seminal fluid of 28 infected individuals was tested and there was no RNA of COVID-19 in the seminal fluid from 8 to 54 days after contamination [49].

In two separate studies, a seminal fluid test and testicular biopsy by real time-polymerase chain reaction (PCR) method reported no presence of COVID-19 in infected patients’ seminal fluid and testis samples [50,51]. Pan et al. demonstrated that there was no COVID-19 in semen fluids 31 days post confirmation of infection with COVID-19 [52]. In another study, Ma and his colleagues evaluated the seminal fluid of 12 patients post COVID-19 infection, and data showed that there was no viral infection in the seminal fluid [53].

### 5.1. Sperm Count

Recently, Hu et al. assessed the effects of COVID-19 infection on sperm quality in recovered patients compared to healthy cases [54]. The results of their study showed that COVID-19 can significantly decrease sperm count in comparison with the control group. It is also reported that sperm count in recovered patients’ after 150 days was similar to the control group. So, at least 5 months recovery time is needed for sperm count to return to normal. Finally, they concluded that the recovery time in COVID-19 patients plays an important role in increasing sperm quality, especially sperm count. Therefore, COVID-19-infected patients after treatment must receive supplemental care to improve sperm count. In another study, the seminal samples of COVID-19-infected patients after recovery were analyzed and compared with healthy men. The results revealed that sperm count was significantly lower in recovered patients than in the control group [55]. Segars et al. demonstrated that the fertility potential is decreased in COVID-19-infected patients and sperm analysis showed reduced sperm count 3 months post COVID-19 infection [56]. Holtmann et al. in their study, evaluated the effects of COVID-19 on sperm parameters, demonstrating that in COVID-19-infected patients, sperm concentration is significantly decreased [49]. In a study published in 2022, sperm count was assessed in COVID-19-recovered patients and there was a significantly decreased sperm count in patients 1 month post recovery in comparison with 3 months, revealing that recovery time is an important factor for sperm count improvement post COVID-19 infection [57].

Guo et al. in 2021 confirmed that sperm count was significantly decreased in COVID-19-infected patients, suggesting that a direct effect of COVID-19 on spermatogenesis may be responsible for sperm count decrease in COVID-19 patients [58]. A study by Li et al. reported that sperm count was decreased in COVID-19 cases, suggesting that elevation of immunological factors, including interleukin-6 (IL-6) and tumor necrosis factor-α (TNF-α), might be responsible for differences in sperm count between COVID-19-infected patients and healthy control individuals [59]. Additionally, another study concluded that the elevation of sperm DNA fragmentation is likely the cause of sperm count decrease in COVID-19-infected patients [53].

### 5.2. Sperm Motility

Several studies investigated the effects of COVID-19 infection on sperm motility and reported decreased sperm motility in patients compared to healthy individuals [56].

Assessing sperm parameters in COVID-19-infected men, Piroozmanesh et al. demonstrated that sperm motility was significantly decreased [60]. Holtman et al. in their study, confirmed a sperm motility decrease in COVID-19 moderately infected patients [49]. Similarly, Li et al. reported that SARS-CoV2 infection can have negative effects on sperm mobility in fertile infected individuals [59]. Recently, Donders et al. reported significantly decreased sperm motility immediately after COVID-19, which was reversed and finally increased 3 months post infection compared to 1 month [57].

Interestingly, in contrast to the above-mentioned papers, Guo et al. analyzed the sperm quality of COVID-19-infected patients and reported that among 23 patients, only in 2 cases was the sperm motility decreased, revealing that in their study population COVID-19 had no negative effect on sperm motility [58]. He et al. in 2021 established that sperm motility was influenced by infection. They reported that sperm motility was significantly decreased during moderate infection in comparison with mild infection or healthy control individuals [61]. The elevation of oxidative stress and damage to sperm DNA are other factors for decreased sperm motility in viral infection cases [62].

### 5.3. Sperm Morphology

Sperm morphology is a vital parameter in the sperm analysis process. Sperm morphology, concerning the size and shape of sperm, is one factor that is examined as part of a semen analysis to evaluate male infertility. Changes in sperm shape may have negative effect on sperm motility, viability and fertility potential. Several factors can generate defects in sperm morphology. Infection is one of these factors. After the COVID-19 pandemic, an important question about whether COVID-19 is a defective factor for sperm morphology has arisen. A recent study investigated the effects of COVID-19 on sperm morphology in three groups, namely infected patients before and after treatment and healthy control males. In the infected patients before treatment, sperm morphology was significantly decreased in comparison with the control group. Furthermore, in treated individuals, sperm morphology was impaired compared to the control group. They concluded that a continuous fever during COVID-19 was an aggravating factor for sperm morphology [63]. A study by Ma et al. demonstrated that the normal morphology of sperm in COVID-19 patients is significantly decreased [53]. In 2021, Falahieh et al. assessed sperm morphology in treated COVID-19 cases after 3 months. The results of their study showed that the percentage of normal sperm morphology increased 3 months after COVID-19 treatment, but based on WHO guidelines, the rate of normal morphology was lower than standard [64]. Maleki et al. showed that sperm morphology can be decreased after SARS-CoV2 infection [65]. Koc and his colleagues [42] investigated sperm parameters of 21 infertile men who were referred for infertility treatment. The results of COVID-19 tests in these cases were positive without requiring hospitalization. This study showed that normal sperm morphology was significantly decreased.

Since COVID-19 can activate several pathways throughout inflammatory responses, it can induce further oxidative stress. This oxidative stress can cause peroxidative impairment to the plasma membrane of sperm. This process can create defects in sperm chromatin and DNA integrity, which leads to abnormal sperm morphology [66]. On the other hand, fever associated with infection constitutes a defective factor for sperm morphology. Other researchers suggested that the level of fever (mild/medium/severe) and its duration (number of days) are two main parameters that can significantly decrease normal sperm morphology parameters [62]. Two separate studies confirmed that high fever levels demonstrated a negative correlation with normal sperm morphology [67,68]. It is speculated that the decrease in normal sperm morphology during COVID-19 can be attributed to fever levels.

### 5.4. Sperm Viability

Sperm viability in COVID-19 infection was investigated in some studies. The results of studies confirmed that sperm viability was decreased in COVID-19 patients [55]. Falahieh et al. assessed sperm analysis of 20 COVID-19 patients 2 weeks and 3 months post infection. They found that sperm viability had no significant difference between two investigation time frames [64]. However, Piroozmanesh and his colleagues compared sperm viability in infected patients and healthy control men, showing that sperm viability was significantly decreased in COVID-19 patients [60].

Several mechanisms have been proposed for the negative effects of COVID-19 on sperm viability. Firstly, there is a hypothesis that influenza SARS-CoV-2 viruses cause oxidative stress that negatively affects sperm viability [69]. Secondly, elevation in inflammatory factors, including cytokines, TNF-α, and interleukin-2, 6, can induce lipid peroxidation in sperm, leading tosperm viability damage [70]. Marin et al. have suggested that COVID-19 can induce ACE2 production, and this is a negative factor for sperm viability in COVID-19 patients [71]. Based on Kopper’s study, COVID-19 may induce apoptosis and lead to sperm viability impairment [72].

## 6. Testicular Changes

Several studies have demonstrated testicular damage and increased orchitis attacks following COVID-19 [73,74]. The function of gonads in SARS-CoV2-infected patients was found to be significantly impaired compared to controls [53]. Bian et al. evaluated testis tissue in COVID-19 patients, revealing considerable damage in seminiferous tubules and a decrease in spermatogenic cell lines [75]. La et al. [76] reported testicular pain in fertile men after COVID-19 infection in both testes, being more severe in the left testis. Li et al. [59] confirmed pathological changes in 6 of 12 patients following COVID-19, where 11 of these patients suffered from orchitis. According to Holtman et al. a mild COVID-19 infection induced the impairment of testicular function [49], whereas a moderate severity infection affected the semen parameters. In contrast to the above-mentioned studies, Song et al. tested testis tissue from men who died from COVID-19 infection, demonstrating no COVID-19-positive cells in testis tissue [48].

Other researchers demonstrated that infections of COVID-19 could have an indirect impact on seminiferous function by affecting the hypothalamic–pituitary–testis axis through immunological and inflammatory responses [77,78]. Xu et al. demonstrated that COVID-19-affected patients showed an increased number of leukocytes infiltrating into the testes, accompanied by an atrophic seminiferous epithelium [73]. The possibility of blood–testis barrier (BTB) breakdown has thus been speculated for COVID-19 patients but still requires experimental validation [79]. Aziz et al. confirmed that COVID-19 can induce immune response and could damage the testis tissue and spermatogenic function due to immune response released mediators [80]. Based on previous observations, COVID-19 can affect CD14 in testes via spermatogonia stem cells [81,82]. A study evaluated that the infiltration and degeneration of CD3+ and CD68+ immune cells in spermatogenic cells can lead to testicular inflammation after COVID-19. Yang et al. showed that viral orchitis was associated with T lymphocyte invasion into the testicular parenchyma in 12 dead patients infected with COVID-19, causing remarkable reproductive tubular damage. They reported that infection with COVID-19 caused histopathological changes in postmortem testicular tissues [83]. This implies that sperm function may be affected by systemic inflammation that leads to increased oxidative stress. During viral infections, fever can impair spermatogenesis, while SARS-CoV2 infection causes testicular apoptosis [84] and disrupts spermatogenesis in general [62]. IgG precipitates were seen in testes infected with COVID-19, and a secondary autoimmune response of extensive IgG production in Sertoli cells and germ cells caused testicular dysfunction [73]. Additionally, IgG has been linked to autoimmune orchitis [85], which might trigger the host’s immune system and cause antibodies to be incorporated into semen fluid [86].

In another study, the researchers hypothesized that an elevation in body temperature is a defective factor for male reproductive tissues [87]. Some data show that afebrile episode due to the viral infection can affect the quality of sperm [88]. This may result in reduced fertility due to testicular damage following COVID-19.

Other mechanisms are linked to ACE2. Changes in ACE2 during infection can damage reproductive tissues. In a paper by Reis et al. [89], ACE2 was reported in reproductive tissues including testis, prostate and epididymis. This is an important factor for male puberty and healthy reproductive function [89]. Multiple reports have demonstrated that COVID-19 causes orchitis, mild lymphocytosis, decreased Leydig cell counts, and the destruction of seminiferous tubules in the testis [90,91,92]. In other words, high ACE2 expression in the testis may facilitate the entrance and colonization of virus, which can have adverse effects on male reproductive functioning [93]. ACE2 is predominantly expressed in the Sertoli and Leydig cells at the transcriptome level in adult human testis [94]. A transcriptome analysis study of human testicular tissues has demonstrated that the TMPRSS2 and ACE2 receptors are prominently expressed in Sertoli cells, Leydig cells, spermatids, and spermatogonia [95].

## 7. Sex Hormones

The negative effects of the viral infection on sex hormones have been validated in several studies [49,96]. In a male hormonal profile, it was confirmed that testosterone is an important hormone in the male reproductive system. Any changes in testosterone level are harmful for male fertility. Decreased levels of testosterone can impair spermatogenesis and fertility potential [97]. Holtmann et al. reported that in COVID-19 patients, the level of sex hormones was decreased significantly [49]. Moreover, Pozzilli et al. [47] revealed that there is a negative correlation between the risk of COVID-19 and the level of the testosterone hormone. The level of testosterone and LH or follicle-stimulating hormone (FSH) is another factor for the diagnosis of male fertility potential. An imbalance in the testosterone/LH/FSH ratio prognosticates defects in male fertility during SARS-CoV2 infection. Furthermore, impaired hypothalamic–pituitary–gonadal axis activity can change hormonal levels and be harmful for healthy reproductive function [98]. Sex hormone concentration was evaluated in 12 patients after infection with COVID-19, reporting a decreased level of testosterone/LH and increased level of LH compared to the controls [53]. Temiz and his colleagues investigated sex hormones in three groups (patients with COVID-19 before and after treatment, and a control group of healthy men). They reported that COVID-19-infected patients before treatment showed a significant decreased level of hormones (testosterone, FSH and LH) in comparison with the control group. Comparison between pre- and post-treatment levels showed that the level of hormones was elevated after treatment, similar to the control group. They concluded that COVID-19 infection has no continuous negative effects on male sex hormones [63]. In the study of Ma et al. the hormonal profile of COVID-19-infected cases was assessed and compared with a healthy control group. The results showed that the level of the LH hormone was significantly higher in COVID-19 cases than controls. Additionally, the testosterone/LH ratio was lower in COVID-19 patients compared to the control group [99]. In another study [100], testosterone levels were decreased, and the LH levels were increased. Consequently, this is an important effect that could impair the balance of the testosterone/LH/FSH ratio. Schroeder et al. [101] found that in COVID-19 men, the levels of FSH and LH were increased, and the level of testosterone decreased. They suggested that these changes in sex hormone levels during the COVID-19 pandemic were due to the activation of the immune system and a down-regulation of hormone production. Interestingly, Cayan S. et al. demonstrated that the average concentration of LH and FSH post COVID-19 infection increases and that the increased levels could be attributed to gonadotropin cell production as a result of early inflammation [102].

The main mechanism that affects hormonal changes during COVID-19 is inflammation in reproductive tissues and the immune response to inflammation. Most androgens are produced by the Leydig cells [103]. Leydig cells were reduced remarkably in 12 patients with COVID-19. It has recently been reported that hypogonadotropic hypogonadism in COVID-19 patients may be related to the replication of COVID-19 in Leydig cells. A recent study by Rastrelli et al. [97] suggests that novel COVID-19 may impair the secretory function of Leydig cells. Moreover, they demonstrated low levels of total dihydrotestosterone in patients with severe COVID-19, speculating that SARS-CoV2 infection may have been related to enhanced inflammation or a cytokine storm, which can lead to decreased testosterone levels [83,97]. A study published in 2019 by Guan X. et al. revealed that testosterone is necessary to preserve male reproductive function, as well as to support the maturation of Sertoli cell and Leydig cell development [104]. It has been reported that TMPRSS2 expression is increased in men upon exposure to androgens [95]. These changes are thought to be the result of Leydig cell damage and the impairment of the steroidogenic pathways [105]. Additionally, psychological stress is an important factor related to elevated levels of oxidative stress in the human body. Increased levels of OS have negative effects on the male reproductive system and can prevent hormone secretion [98]. COVID-19 is characterized as a high-stress disease, and this psychological condition may be responsible for the hormone impairment in COVID-19 men [106]. The key studies of the effects of COVID-19 infection and vaccination on male fertility parameters are shown in Table 1.

## 8. Conclusions

Recent multi-center research during the outbreak indicates that male fertility might be impaired as a possible consequence of COVID-19. Hence, more attention should be paid to the effect of COVID-19 infection on the male reproductive system. So far, the clinical trials have shown that there is no defective effect of COVID-19 vaccines on male fertility potential. The lack of information regarding the effect of SARS-CoV-2 infection on male fertility represents a weakness for the present understanding of COVID-19. Therefore, we suggest that the medical community should provide assurance to the population about the safety of vaccines for male reproductive function. As COVID-19 infection has demonstrated a clear correlation to male infertility, it is suggested that men should consider preserving their fertility by cryo-preserving their spermatozoa. Vaccination is an important way to prevent the negative effects of COVID-19 on male fertility.

## Figures and Tables

**Figure 1 vaccines-10-01982-f001:**
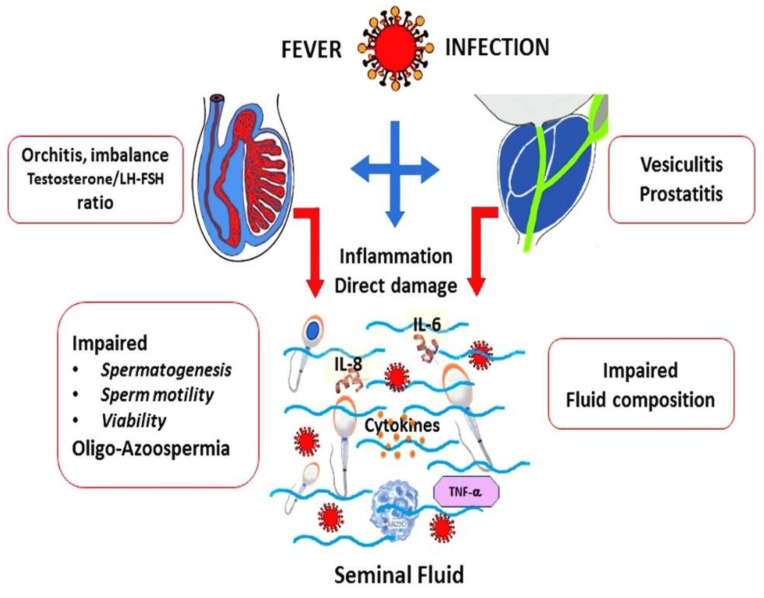
Once infection begins, the virus can affect the testes and prostate through the interaction with angiotensin I-converting enzyme 2 (ACE2) receptors and transmembrane serine protease. However, this complication does not occur in every patient with COVID-19 disease, being more likely in predisposed individuals with lengthy infection or fever. As the virus moves into their reproductive organs, seminal fluid and sperm characteristics are modified. Infection may also cause direct seminal fluid inflammation and sperm damage, with polymorphonucleate and lymphocyte activation, interleukins (IL-6, IL-8, others) and cytokine release, tumor necrosis factor (TNF-α), etc. Most authors reported these as transient complications (see text).

**Table 1 vaccines-10-01982-t001:** COVID-19 vaccination and male reproductive potential. Summarization of some key studies of the effects of COVID-19 infection and vaccination on male fertility parameters.

First Name	Type of Study	Finding and Conclusion
**Seminal Fluid**		
Olana et al. (2022) [34]	Prospective study	Inflammatory factors including IL-6 did not increase after vaccination in human seminal fluid.
Gacci et al. (2021) [43]	Cohort study	IL-8 was detected in seminal fluid of COVID-19 patients.
Li et al. (2020) [44]	Cohort study	Semen can shed the virus during the recovery phase, even though COVID-19 does not affect the semen of all men who have the virus.
Maya et al. (2020) [45]	Commentary	Coronavirus is detectable in acute infected patients after treatment and in the recovery period. They also discussed that COVID-19 is detectable in patients who have no infection symptoms.
Song et al. (2020) [48]	Letter to the Editor	They cannot identify the presence of COVID-19 in the seminal fluid of male patients after recovery.
Holtman et al. (2020) [49]	Cohort study	There was no RNA of COVID-19 in seminal fluid 8 to 54 days after contamination.
Paoli et al. (2020) [51]	Case report	Negative presence of COVID-19 in infected patient’s seminal fluid and testis sample.
Pan et al. (2020) [52]	Observational, cross-sectional study	No SARS-CoV2 was found in semen fluids 31 days after confirming their infection with COVID-19.
**Sperm Count**		
Zhu et al. (2022) [35]	Cohort study	There was no decrease in sperm count and volume after COVID-19 vaccination in comparison with before vaccination.
Barda et al. (2022) [36]	Cohort study	Sperm concentration count significantly increased after the second vaccine.
Chatzimeletiou et al. (2022) [37]	Case report	The sperm count significantly increased from first to third vaccine dose.
Holtman et al. (2020) [49]	Cohort study	In COVID-19 patients, sperm concentration significantly decreased.
Ma et al. (2021) [53]	Case control study	Elevation of sperm DNA fragmentation is the cause of sperm count decrease in COVID-19 men.
Hu et al. (2022) [54]	Case control study	COVID-19 can significantly decrease sperm count in comparison with control group. They also reported sperm count in recovered patients after 150 days was similar to control group.
Ruan et al. (2021) [55]	Cohort study	Sperm count was significantly lower in recovered patients than control group.
Segars et al. (2020) [56]	Review study	Sperm analysis showed reduced sperm count 3 months after COVID-19.
Donders et al. (2020) [57]	Cohort study	There was a significant decreased sperm count in patients 1month after recovery in comparison with 3 months after recovery.
Guo et al. (2021) [58]	Cohort study	COVID-19 direct attack on sperm is responsible for sperm count decrease in COVID-19 patients.
Li et al. (2020) [59]	Observational study	Elevation of immunological factors including interleukin-6 (IL-6) and tumor necrosis factor-α (TNF-α) are responsible for the difference in sperm count between COVID-19 patients and healthy control men.
**Sperm Motility**		
Olana et al. (2022) [34]	Prospective study	There was no significant difference in OS level and sperm motility before and after vaccination.
Zhu et al. (2022) [35]	Cohort study	COVID-19 vaccination no had negative effects on sperm motility.
Chatzimeletiou et al. (2022) [37]	Case report	In ankylosing spondylitis patients, COVID-19 vaccination can improve sperm motility after second dose.
Holtman et al. (2020) [49]	Cohort study	Sperm motility decreased in COVID-19 moderately infected patients.
Donders et al. (2020) [57]	Cohort study	Sperm motility significantly decreased immediately after COVID-19, and over time (3 months after infection), sperm motility was repaired and increased compared to 1 month after infection.
Guo et al. (2021) [58]	Cohort study	COVID-19 no had negative effect on sperm motility.
Li et al. (2021) [59]	Observational study	COVID-19 had negative effects on sperm motility in fertile infected men.
Piroozmanesh et al. (2022) [60]	Case control study	Sperm motility significantly decreased in COVID-19 patients.
He et al. (2021) [61]	Systematic review	Sperm motility significantly decreased in moderate infection in comparison with mild infection or healthy control cases.
**Sperm Morphology**		
Chatzimeletiou et al. (2022) [37]	Case report	COVID-19 vaccination had no negative effects on sperm morphology, although the shape of sperm in head and neck after ankylosing spondylitis showed more defects. However, vaccination had positive effects on sperm morphology.
Kumar et al. (2021) [38]	Letter to the Editor	There was no scientific evidence to confirm the damaging effects of COVID-19 vaccination on sperm parameters.
Koc et al. (2021) [42]	Research study	Normal sperm morphology in COVID-19 cases significantly decreased.
Ma et al. (2021) [53]	Case control study	Normal morphology of sperm in COVID-19 patients significantly decreased.
Temiz et al. (2021) [63]	A cross-sectional, pilot study	Infected patients before sperm morphology treatment significantly decreased in comparison with control group.
Falahieh et al. (2021) [64]	Prospective study	Percentage of normal sperm morphology increased 3 months after COVID-19 treatment but based on WHO guideline, the rate of normal morphology was lower than standard.
Maleki et al. (2021) [65]	Cohort study	Sperm morphology can decrease after COVID-19.
**Sperm Viability**		
Ruan et al. (2021) [55]	Cohort study	Sperm viability decreased in COVID-19 patients.
Piroozmanesh et al. (2022) [60]	Case control study	Sperm viability significantly decreased in COVID-19 patients.
Falahieh et al. (2021) [64]	Prospective study	Sperm viability showed no significant difference between 2 weeks and 3 months after infection.
Marin et al. (2021) [71]	Letter to the Editor	Coronavirus can induce ACE2 production, and this is a negative factor on sperm viability in COVID-19 patients.
**Testis Tissue**		
Stanley et al. (2020) [12]	Descriptive study	COVID-19 used Basigin (BSG) and protease Cathepsin L (CTSL) receptors to enter Leydig cells.
Kumar et al. (2021) [38]	Letter to the Editor	Rate of orchitis and epididymitis were significantly lower in vaccinated cases compared to non-vaccinated ones. They also concluded a single dose of COVID-19 vaccine can significantly decrease the risk of inflammation in testis.
Song et al. (2020) [48]	Letter to the Editor	There were no COVID-19-positive cells in testis tissue.
Holtman et al. (2020) [49]	Cohort study	Mild infection of COVID-19 impairs testicular function.
Ma et al. (2021) [53]	Case control study	The function of gonads in COVID-19-infected patients was significantly lower than control group.
Li et al. (2020) [59]	Observational study	Pathological changes observed in 6 of 12 expired patients after COVID-19 infection.
Bian et al. (2020) [75]	National science review	Considerable impairment was detected in seminiferous tubules and a decrease in spermatogenic cell lines of COVID-19 patients.
LaMarca et al. (2020) [76]	Review study	In COVID-19-infected cases, testicular pain in both sides by severity in left testis was observed.
Abobaker et al. (2021) [77]	Letter to the Editor	COVID-19 could have an indirect impact on seminiferous function by affecting the hypothalamic–pituitary–testis axis through immunological and inflammatory responses.
Olaniyan et al. (2021) [79]	Review study	BTB breakdown has thus been speculated for COVID-19 patients but still requires experimental validation.
Ulrish et al. (2020) [81]	Review study	COVID-19 infection can affect CD14 in testes via spermatogonia stem cells.
Yang et al. (2020) [83]	Research study	Infection with COVID-19 has been shown to cause histopathological changes in postmortem testicular tissues.
Teixeira et al. (2021) [91]	Mini review	COVID-19 causes orchitis, mild lymphocytosis, decreased Leydig cell counts, and destruction of seminiferous tubules in the testis.
Lukassen et al. (2020) [93]	Research study	High ACE2 expression in the testis may facilitate entrance and colonization of virus, which can have adverse effects on male reproductive function.
**Sex Hormones**		
Adamyan et al. (2022) [33]	Review study	No negative effects of the COVID-19 vaccine on the level of testosterone, FSH and LH hormone after vaccination.
Pozzilli et al. (2020) [47]	Mini review	There was a negative correlation between the risk of COVID-19 and level of testosterone hormone.
Holtman et al. (2020) [49]	Cohort study	In COVID-19 patients, the level of sex hormones decreased significantly.
Ma et al. (2021) [53]	Case control study	Decreased level of testosterone/LH and increased level of LH in COVID-19 patients compared to the controls.
Temiz et al. (2021) [63]	A cross-sectional, pilot study	Infected patients with COVID-19 before treatment showed significant decreased level of hormones (testosterone, FSH and LH) in comparison with control group. Comparison between before and after treatment showed the level of hormones elevated after treatment and was similar to control group.
Rastrelli et al. (2021) [97]	Research study	Patients with severe COVID-19 had low levels of total dihydrotestosterone.
Li et al. (2020) [98]	Review study	Imbalance of testosterone/LH/FSH ratio is prognostication of defects in male fertility during COVID-19.
Ma et al. (2020) [99]	Research study	The level of LH hormone was significantly higher in COVID-19 cases than control. Additionally, the testosterone/LH ratio was lower in COVID-19 patients than control.
Jose et al. (2020) [100]	Systematic review	Level of sex hormones during COVID-19 showed decrease in the level of testosterone and increased in the level of LH.
Cayan et al. (2020) [102]	Cohort study	Concentration of LH and FSH after COVID-19 increases.

## Data Availability

Not applicable.

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
