# Peer review of "The Effect of Long COVID-19 Infection and Vaccination on Male Fertility; A Narrative Review"

_vaccines, 2022, doi:10.3390/vaccines10121982_

Round 1

Reviewer 1 Report

Dear Editor,

Thank you for taking me into consideration for the review process of the article titled “The effect of long COVID-19 infection and vaccination on male 2 fertility; a narrative review”. I read it attentively and have the following comments:

1.     Several lesser mistakes were identified during the revision process (e.g., line 144: “How a virus can cause damage […]?” [Correct phrasing: How can a virus cause damage]; line 146 and following: multiple instances of “virus” [singular] used instead of “viruses” [plural]; line 156: “the inflammation chain can create damage to [the] body”). A revision of the text’s English language is advised.

2.     The Search methods sections should report the full research string employed for searching the databases. Selected databases should also be specified and, although this is not a systematic review, a full count of included studies would be useful for the reader.

3.     The third section is called “SARS-CoV-19”. However, the correct name for the pathogen is SARS-CoV-2. The authors should change it as needed, while also looking for other possible instances of similar mistakes throughout the text (e.g., line 132: “COVID 19” [hyphen is missing]). Coronavirus is also frequently misspelled (e.g., line 216-217 “corona virus”).

4.     In section 4, pre-licensure clinical trials should be reported as source of preliminary evidence.

5.     I find the study methods wanting. In fact, no single criterion is defined in order to account for “fertility effects on males”. Studies are considered both evaluating sperm count and ignoring it, the latter exclusively relying on information about the subjects’ vaccination status, nationality and occupation. Since male infertility is assessable reliably only via quantitative and qualitative sperm examination, studies which do not take these tests into consideration should not be included by the review, or at least should be discussed separately to avoid confusion.

6.     Studies included by the review are often low-numerosity ones. This indicates a significant gap of knowledge, thus representing a limitation for the review itself. The Conclusion section should therefore mention that the lack of information regarding SARS-CoV-2 infection’s effect on male fertility represents a weakness for present understanding of COVID-19.

Author Response

//

  1. Several lesser mistakes were identified during the revision process (e.g., line 144: “How a virus can cause damage […]?” [Correct phrasing: How can a virus cause damage]; line 146 and following: multiple instances of “virus” [singular] used instead of “viruses” [plural]; line 156: “the inflammation chain can create damage to [the] body”). A revision of the text’s English language is advised.
  2. The Search methods sections should report the full research string employed for searching the databases. Selected databases should also be specified and, although this is not a systematic review, a full count of included studies would be useful for the reader.
  3. The third section is called “SARS-CoV-19”. However, the correct name for the pathogen is SARS-CoV-2. The authors should change it as needed, while also looking for other possible instances of similar mistakes throughout the text (e.g., line 132: “COVID 19” [hyphen is missing]). Coronavirus is also frequently misspelled (e.g., line 216-217 “corona virus”).
  4. In section 4, pre-licensure clinical trials should be reported as source of preliminary evidence.
  5. I find the study methods wanting. In fact, no single criterion is defined in order to account for “fertility effects on males”. Studies are considered both evaluating sperm count and ignoring it, the latter exclusively relying on information about the subjects’ vaccination status, nationality and occupation. Since male infertility is assessable reliably only via quantitative and qualitative sperm examination, studies which do not take these tests into consideration should not be included by the review, or at least should be discussed separately to avoid confusion.
  6. Studies included by the review are often low-numerosity ones. This indicates a significant gap of knowledge, thus representing a limitation for the review itself. The Conclusion section should therefore mention that the lack of information regarding SARS-CoV-2 infection’s effect on male fertility represents a weakness for present understanding of COVID-Answers

//

Points 1,2,3,4,6: We thank the reviewer. We have corrected the mistakes and the study methods according to the reviewer's suggestions. All corrections are shown in the text in red color

Point 5: We thank  the reviewer for the so useful suggestions

Studies reporting the effect of either SARS-CoV2 infection or COVID-19 vaccination on qualitative factors such as demographic parameters and quantitative markers such seminal fluid quality, sperm count, sperm motility, sperm morphology elements, sperm viability, testicular and sex hormone function were analyzed.   The studies concerning the effect of COVID -19 vaccination on quantitative markers  have been removed to paragraph 4

Reviewer 2 Report

The paper is interesting and well written. I suggest to discuss the role of IL-31/IL-33 axis, vitamin D and microbiota in immune responses including vaccinations (see and add as references papers by Murdaca et al concerning tese topics).

Author Response

//

The paper is interesting and well written. I suggest to discuss the role of IL-31/IL-33 axis, vitamin D and microbiota in immune responses including vaccinations (see and add as references papers by Murdaca et al concerning tese topics).ANSWERS

ANSWERS:

We are much obliged to reviewer 2 for the so useful comments and advise. We have searched very careful the scientific literature and we could not find any published paper relating COVID-19 vaccination male infertility with IL-31/IL-33 axis, vitamin D, microbiota.

Round 2

Reviewer 1 Report

Authors right addressed my previous concerns.